

# Comparative spigot ontogeny across the spider tree of life

Rachael E. Alfaro[1], Charles E. Griswold[2] and Kelly B. Miller[1]

[1] Museum of Southwestern Biology, Division of Arthropods, University of New Mexico, Albuquerque, NM, United States of America

[2] Entomology, California Academy of Sciences, San Francisco, CA, United States of America

## ABSTRACT

Spiders are well known for their silk and its varying use across taxa. Very few studies have examined the silk spigot ontogeny of the entire spinning field of a spider. Historically the spider phylogeny was based on morphological data and behavioral data associated with silk. Recent phylogenomics studies have shifted major paradigms in our understanding of silk use evolution, reordering phylogenetic relationships that were once thought to be monophyletic. Considering this, we explored spigot ontogeny in 22 species, including *Dolomedes tenebrosus* and *Hogna carolinensis,* reported here for the first time. This is the first study of its kind and the first to incorporate the Araneae Tree of Life. After rigorous testing for phylogenetic signal and model fit, we performed 60 phylogenetic generalized least squares analyses on adult female and second instar spigot morphology. Six analyses had significant correlation coefficients, suggesting that instar, strategy, and spigot variety are good predictors of spigot number in spiders, after correcting for bias of shared evolutionary history. We performed ancestral character estimation of singular, fiber producing spigots on the posterior lateral spinneret whose potential homology has long been debated. We found that the ancestral root of our phylogram of 22 species, with the addition of five additional cribellate and ecribellate lineages, was more likely to have either none or a modified spigot rather than a pseudoflagelliform gland spigot or a flagelliform spigot. This spigot ontogeny approach is novel and we can build on our efforts from this study by growing the dataset to include deeper taxon sampling and working towards the capability to incorporate full ontogeny in the analysis.

## INTRODUCTION

Silk is the trait most commonly associated with spiders. Silk is produced by glands that service spigots on specialized appendages called the spinnerets. Spinnerets are a distinguishable synapomorphy of Araneae (*Coddington, 1989*; *Platnick, 1990*; *Platnick & Griswold, 1991*; *Griswold et al., 2005*; *Wheeler et al., 2016*). The morphology of the spinnerets and the silk spigots they possess provides an advantage enabling spiders to create simple to complex silk structures from sheet-webs to tangle webs (*Selden, Shear & Sutton, 2008*). The evolutionary history of spiders has long been explored in the context of silk evolution. With the arrival of molecular phylogenetics and phylogenomics studies, our understanding of spider systematics has changed drastically from the formerly well accepted

Corresponding author
Rachael E. Alfaro, mallis@unm.edu

hypotheses based on morphological and behavioral traits (*Platnick, 1977*; *Griswold et al., 2005*; *Bond et al., 2014*; *Fernández, Hormiga & Giribet, 2014*; *Garrison et al., 2016*; *Wheeler et al., 2016*).

These recent updates have led to a paradigm shift in our perception of silk use evolution. The most dramatic changes have occurred in the ''Orbiculariae'' where data mainly from orb web weaving behavior provided weak corroborating evidence of monophyly, whereas contradictory data were lacking. Specifically, the orb web was considered a key adaptation in spider evolution (*Bond & Opell, 1998*). However, despite previous support for this hypothesis through morphological and behavioral data (*Coddington, 1986*; *Coddington, 1990*; *Hormiga & Griswold, 2014*), the monophyly of Orbiculariae (cribellate Deinopoidea + viscous Araneoidea) has now been rejected based on thorough molecular and phylogenomics studies (*Dimitrov et al., 2012*; *Bond et al., 2014*; *Fernández, Hormiga & Giribet, 2014*; *Garrison et al., 2016*; *Wheeler et al., 2016*). The former ''Orbiculariae'' members Deinopoidea (cribellate orb builders) are now closer to the RTA clade (includes wolf spiders and jumping spiders) rather than to the Araneoidea (sticky-silk orb weavers) and Deinopoidea may not even be monophyletic (*Garrison et al., 2016*; *Wheeler et al., 2016*).

Some studies of silk evolution have used web ontogeny as a tool to reconstruct ancestral web conditions or plesiomorphic traits in silk use. Studies many from to the Araneoidea (*Robinson & Lubin, 1979*; *Eberhard, 1985*; *Eberhard, 1986*; *Barrantes & Madrigal-Brenes, 2008*; *Barrantes & Eberhard, 2010*), suggested that early instar webs and behavior resembled possible ancestral states. In studies of both *Tengella perfuga* F. Dahl (1901) and *Tengella radiata* (W. Kulczyński, 1909), early instar webs resembled simple sheet webs rather than the complex funnel structures lined with cribellate silk observed in adults (*Barrantes & Madrigal-Brenes, 2008*; *Mallis & Miller, 2017*). This may be the ancestral condition for this lineage. However, *Mallis & Miller (2017)* observed *T. perfuga* lay down cribellate silk in an orb-like spiral within the horizontal sheet. This observation makes sense, considering recent phylogenomics revisions and results of the new Araneae Tree of Life (AToL) (*Wheeler et al., 2016*) project, where the sister group to the RTA clade is now hypothesized to be the cribellate orb weavers of the Uloboridae (*Bond et al., 2014*; *Fernández, Hormiga & Giribet, 2014*; *Garrison et al., 2016*; *Wheeler et al., 2016*).

One of the current hypotheses about silk evolution is that there is an adaptive tradeoff between fecundity and silk use that is driving spider evolution and where more recently derived clades have lost silk as a foraging tool (*Blackledge, Coddington & Agnarsson, 2009*). Energy metabolism in a spider species is related to natural history traits such as foraging activity level and courtship behaviors (*Anderson, 1970*; *Prestwich, 1977*; *Anderson & Prestwich, 1982*; *Prestwich, 1983*; *Anderson, 1996*). Foraging activity level is also tied to the type of silk used and web building or non-web building. Cribellate silk has been historically viewed as a plesiomorphic trait in spiders, which requires the development and use of the cribellum (a plate derived from the ancestral anterior median spinnerets) and the calamistrum (a setal comb on leg IV to pull the silk fibers out) (*Hawthorn & Opell, 2002*; *Blackledge, Coddington & Agnarsson, 2009*; *Pechmann et al., 2010*). It is possible that that trend toward higher fecundity that is observed in orb-weavers and non-web builders, when compared to cribellate silk users supports the hypothesis of ''adaptive escape'' from
metabolically costly cribellate silk production and represents increased resource allocation to reproduction in spiders that produce viscous silk or that are non-web building altogether (*Blackledge, Coddington & Agnarsson, 2009*).

We wanted to explore the potential correlations of foraging strategy and silk use and did so in the context of the silk spigots themselves. Each spigot is serviced by a specific gland and each type of silk serves a different purpose. This has been most well explored in Araneidae (*Coddington, 1989*; *Yu & Coddington, 1990*; *Townley & Tillinghast, 2009*; *Garb, 2013*). There must be a caveat that in most cases the actual gland has not been observed and the spigot type is assigned a gland type name based on inferential evidence such as position, fine structure and ontogeny. Most of the Araneomorphae spiders possess five types of spigots (*Griswold et al., 2005*). These are the (1) major ampullate gland spigots (MAP) on the anterior lateral spinneret (ALS), which produces dragline silk and structural silk for orb webs; (2) piriform gland spigots (PI) on the ALS that produce silk that is used to attach the dragline to a substrate surface; (3) minor ampullate gland spigots (mAP) on the posterior median spinneret (PMS), whose silk is used as a temporary scaffolding for the spiral in the orb web and whose purpose in non-web builders is not yet defined; (4) aciniform gland spigots (AC) on the PMS and (5) aciniform gland spigots on the PLS, which produce silk used in prey wrapping and lining egg sacs, as well as the sheet portions in non-orb webs. Another two types appear in the adult female instar of the Entelegynae, i.e., spiders with sclerotized, flow-through female genitalia (*Griswold & Ramírez, 2017*) and their closest relatives (Austrochiloidea, Palpimanoidea and Leptonetidae), which together form "CY spigot clade" (*Wheeler et al., 2016*). These are (6) cylindrical (=tubuliform) gland spigots (CY) on the PMS and (7) cylindrical gland spigots on the PLS which are female specific and produce fibers that form the egg sac (Fig. 1, and see Fig. 1 in *Garb, 2013*). Because they appear on two different and morphologically distinct spinnerets, we treat the AC and CY gland spigots on the PMS and PLS separately. Araneoids also possess flagelliform (FL) gland and aggregate (AG) gland spigots, which produce the sticky capture spiral in orb webs (*Yu & Coddington, 1990*; *Townley & Tillinghast, 2009*; *Garb, 2013*). The former Deinopoids do not possess flagelliform or aggregate gland spigots but instead possess a cribellum, paracribellar spigots on the PMS (which attach the cribellate silk to its axial line), and the pseudoflagelliform (PF) gland spigot (a possible MS spigot homologue) that produces the axial lines of cribellate fibers (*Hajer, 1991*; *Eberhard & Pereira, 1993*). These cribellar fibrils serve as a prey-capture mechanism, rather than the viscous capture spiral of orb webs. Other, more recently derived cribellate spiders, such as the zoropsid *T. perfuga*, possess a modified spigot on the PLS which is thought to produce the axial line (RE Alfaro, 2017, unpublished data). Zoropsids do not possess paracribellar spigots on the PMS. However, in *T. perfuga*, the modified spigot is flanked by two smaller, unknown spigots whose function is currently undetermined (RE Alfaro, 2017, unpublished data; also observed in *T. radiata* in *Griswold et al., 2005*). We will refer to these flanking spigots of the modified spigot as 'flankers'.

Adult spider silk spigot morphology has been used as a morphological character system in many phylogenetic studies (*Coddington, 1989*; *Platnick, 1990*; *Platnick & Griswold, 1991*; *Griswold et al., 2005*; *Ramírez, 2014*). However, few studies have explored or incorporated

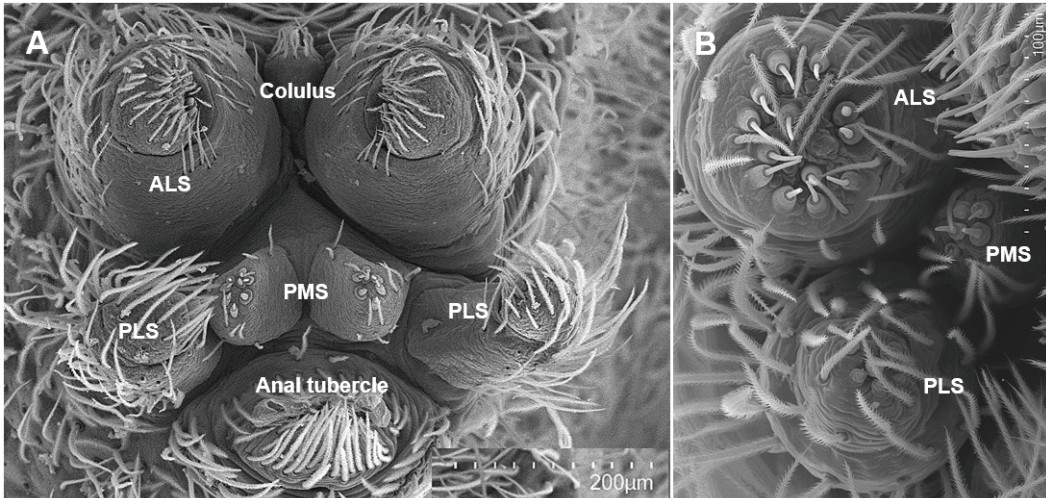

**Figure 1** **Scanning electron micrographs of a spider spinning apparatus morphology.** Spinning fields of *Dolomedes tenebrosus* (A, whole field, 7th instar) and *Hogna carolinensis* (B, right field, 4th instar). ALS, Anterior lateral spinnerets; PMS, Posterior median spinnerets; PLS, Poster lateral spinnerets.

the ontogeny of the whole spinning field (*Wąsowska, 1977*; *Yu & Coddington, 1990*; *Hajer, 1991*; *Townley & Tillinghast, 2009*; *Dolejš et al., 2014*; R Carlson & CE Griswold, 1996, unpublished data). By incorporating the ontogeny of the spinning field, we can observe when activity patterns shift in one adult sex, as well as the appearance or disappearance of spigots, or whether certain spigots increase in number with each molt to a new instar (*Townley & Tillinghast, 2009*; *Mallis & Miller, 2017*; RE Alfaro, 2017, unpublished data).

Here, we report for the first time the ontogeny of the spinning field of *Dolomedes tenebrosus* N.M. Hentz (1884), a fishing spider (Pisauridae), and *Hogna carolinensis* (C.A. Walckenaer, 1805), a wolf spider (Lycosidae). We also use the recently published AToL (*Wheeler et al., 2016*) to conduct the first statistical phylogenetic comparative study of spigots and silk use in spiders. We pooled these two datasets (*Dolomedes* and *Hogna*), as well as our previous study of the cribellate zoropsid *Tengella perfuga*, along with five previously published studies (*Wąsowska, 1977*; *Yu & Coddington, 1990*; *Hajer, 1991*; *Townley & Tillinghast, 2009*; *Dolejš et al., 2014*; RE Alfaro, 2017, unpublished data) and one unpublished dataset representing *Phyxelida tanganensis* (Simon and Fage) (Phyxelididae) (R Carlson & CE Griswold, 1996, unpublished data).

The main objective of our study is to explore trends in silk evolution using phylogenetic comparative methods focusing on correlations between predictor variables such as foraging strategy, and response variables such as the average number of aciniform spigots on the PMS. Therefore, considering the spigot ontogeny of several species across the phylogeny with various foraging strategies and types of silk expressed, and the new Araneae Tree of Life (*Wheeler et al., 2016*), we may be able to disentangle the variables that are correlated such as spigot number, type, foraging strategy and determine what may be driving silk use evolution after correcting for shared evolutionary history (phylogenetic correction).

The four questions guiding our approach are: (1) Is there a relationship between foraging strategy (web vs. non-web) or specific foraging strategies (i.e., ambush, active, sit & wait, etc.) and the number of certain silk spigots in spiders? (2) Is the overall diversity of spigots possessed by a species correlated with spigot number? (3) Does ontogeny have an effect on the number of spigots? and (4) Are there homologous spigots across taxa, particularly the singular, fiber producing spigot (MS, FL, PF) the on the PLS?

## MATERIALS & METHODS

### Spider husbandry

Rearing conditions and lab colony maintenance for *Hogna carolinensis* and *Dolomedes tenebrosus* follow *Mallis & Miller (2017)*. The founding female *H. carolinensis* was collected in Bernalillo County, New Mexico (D. Lightfoot, 10-Sept-2014) carrying second instar spiderlings on her abdomen. A gravid *D. tenebrosus* with an egg sac was sent to us from Bedford County, Virginia (K. Benson, 29-June-2014). When possible, two to three replicates of each instar were randomly sacrificed for scanning electron microscopy (SEM) imaging to account for variation among individuals of the same instar. After developing through the first few instars of both species, colony survival strongly declined. Subsequently, single samples were collected at each instar for *D. tenebrosus*, while after the seventh instar in *H. carolinensis*, a single female was followed to adulthood (twelfth instar). The founding females, as well as instar vouchers, were deposited at the Museum of Southwestern Biology, Division of Arthropods (MSBA 50049–50070).

### SEM preparation, imaging and spigot mapping

We dissected in 100% EtOH, and then critical point dried using a Denton DCP-1 critical point dryer, and mounted specimens of each instar for *D. tenebrosus* and *H. carolinensis* onto standard SEM stubs using a combination of copper tape and non-conductive glue at California Academy of Sciences (CAS). Finally, all SEM stubs were coated in gold/palladium using a Cressington Sputter coater 108 (6002, 6006 series) that used Argon gas to facilitate coating. At CAS, we obtained SEM images on the Hitachi SU-3500 scanning electron microscope. Up to 20 views of each instar spinning field for both species were captured, covering all spinnerets for two replicates of each instar. These SEMs were used to create spigot maps which were translated into a spigot ontogeny dataset for each species (methods outlined in *Coddington (1989)* and *Griswold et al. (2005)*; see Table 1). Spigot mapping allows for notation of type, functionality and placement of silk spigots on each spinneret, as well as tracking the growth of the spinning fields from instar to instar and the appearance of interesting spigot formations.

### Spigot ontogeny datasets

We compiled a large spigot ontogeny dataset of 22 species comprising thirteen spider families using previously published studies and unpublished datasets (Table 2; *Wąsowska, 1977*; *Yu & Coddington, 1990*; *Hajer, 1991*; *Townley & Tillinghast, 2009*; *Dolejš et al., 2014*; R Carlson & CE Griswold, 1996, unpublished data). Three of those sources came from our lab colonies not only for *D. tenebrosus* and *H. carolinensis*, as reported here but also
**Table 1  Full spigot ontogeny of *Dolomedes tenebrosus* and *Hogna carolinensis*.**  Numbers of functional spigots by spinneret for each instar. Pre-spigots are noted for those whose functionality is determined.

| | | Spinneret | | | | | | | |
| | | ALS | | PMS | | | PLS | | |
| Species | Instar | MAP | PI | mAP | AC | CY | AC | CY | Modified |
|---|---|---|---|---|---|---|---|---|---|
| *Dolomedes tenebrosus* | 2 | 2 | 6 | 2 | 4 | 0 | 4 | 0 | 0 |
| *Hogna carolinensis* | 2 | 2 | 4 | 2 | 4 | 0 | 7 | 0 | 0 |
| *Dolomedes tenebrosus* | 3 | 2 | 9 | 2 | 4 | 0 | 6 | 0 | 0 |
| *Hogna carolinensis* | 3 | 2 | 7 | 2 | 6 | 0 | 9 | 0 | 0 |
| *Dolomedes tenebrosus* | 4 | 2 | 9 | 2 | 5 | 0 | 6 | 0 | 0 |
| *Hogna carolinensis* | 4 | 2 | 11 | 2 | 3 | 0 | 3 | 0 | 0 |
| *Dolomedes tenebrosus* | 5 | 2 | 14 | 2 | 5 | 0 | 6 | 0 | 0 |
| *Hogna carolinensis* | 5 | 2 | 13 | 2 | 3 | 0 | 3 | 0 | 0 |
| *Dolomedes tenebrosus* | 6 | 2 | 16 | 2 | 5 | 0 | 9 | 0 | 0 |
| *Hogna carolinensis* | 6 | 2 | 17 | 2 | 6 | 0 | 7 | 0 | 0 |
| *Dolomedes tenebrosus* | 7 | 2 | 18 | 2 | 6 | 0 | 8 | 0 | 0 |
| *Hogna carolinensis* | 7 | 2 | 27 | 2 | 4 | 0 | 7 | 0 | 0 |
| *Dolomedes tenebrosus* | 8 | 2 | 27 | 2 | 8 | 0 | 9 | 0 | 0 |
| *Dolomedes tenebrosus* | 9 | 2 | 57 | 2 | 8 | 0 | 15 | 0 | 0 |
| *Dolomedes tenebrosus* | 10 antepen- ♀ | 2 | 52 | 2 | 8 | 0 | 10 | 0 | 0 |
| *Dolomedes tenebrosus* | 11 pen- ♂ | 2 | 75 | 2 | 9 | 0 | 14 | 0 | 0 |
| *Dolomedes tenebrosus* | 12 pen- ♀ | 2 | 133 | 2 | 30 | Pre1 | 42 | Pre3 | Pre1 |
| *Dolomedes tenebrosus* | 12 ♂ | 1 | 54 | 1 | 7 | 0 | 13 | 0 | 0 |
| *Hogna carolinensis* | 12 ♀ | 2 | 122 | 2 | 82 | 10 | 43 | 1 | 1[a] |
| *Dolomedes tenebrosus* | 13 ♀ | 2 | 107 | 2 | 15 | 32 | 24 | 28 | 1 |

**Notes.**
[a]Indicates a tentative identification which requires more replicates to confirm.

the cribellate zoropsid, *Tengella perfuga* (RE Alfaro, 2017, unpublished data). The dataset included the appearance, type, and number of specific spigots on each spinneret (Fig. 1, Table S1). While some studies reported tartipores (remnant structures of spigots from previous instar spigots) and others the presence of nubbins (non-functioning spigots), all studies reported the number of the seven common (shared) spigot types across all the species of this study: 1. MAP gland spigots, 2. PI gland spigots, 3. mAP gland spigots, 4. AC gland spigots on the PMS, 5. CY gland spigots on the PMS, 6. AC gland spigots on the PLS and 7. CY gland spigots on the PLS. We standardized the final data set to include these 'standard 7' spigots (Table S1).

## Variables of interest

From the standardized datasets, we selected five independent or predictor variables for phylogenetic comparative analyses, and chose twelve dependent or response variables for the final analyses. The independent variables were (Table 2):

1. Strategy: Foraging strategy scored as 0: web builder or 1: non-web builder.
2. Specific: Specific foraging strategies, scored as 1: sit & wait, 1.5: ambush, 2: sit & pursue, 2.5: stalking, 3: active, 4: sheet web, 4.5: funnel web or 5: orb web. Sit & pursue, while

**Table 2 Species included in PGLS analyses by spider family.** Twenty-two species across thirteen spider families were used in the PGLS analyses. The scoring for the five independent variables, from Foraging Strategy to Maximum Number of Instars is reported here. 'Strategy' delineates web (0) or no web usage (1). 'Specific' refers to the type of prey pursuit (1–3) or web (4–5) used by each species. 'Silk' quantifies the main type of silk used by each species from aciniform (1) to viscous (4). 'Type' is a numerical designation for the variety of silk spigot types each species possesses from the 'standard 7' (1) to standard 7 + paracribellar + cribellum + pseudoflagelliform (4). Finally, 'Instar' refers to the maximum number of instars reported for adult females of each species.

| Family | Species | Strategy | Specific | Silk | Type | Instar |
|---|---|---|---|---|---|---|
| Philodromidae | *Tibellus oblongus* | 1 | 2 | 1 | 1 | 6 |
| Thomisidae | *Xysticus cristatus* | 1 | 1.5 | 1 | 1 | 6 |
| Lycosidae | *Xerolycosa nemoralis* | 1 | 3 | 1 | 1 | 10 |
| | *Pardosa lugubris* | 1 | 3 | 1 | 1 | 7 |
| | *Pardosa amentata* | 1 | 3 | 1 | 1 | 9 |
| | *Hogna carolinensis* | 1 | 2 | 1 | 2 | 12 |
| | *Arctosa lutetiana* | 1 | 1.5 | 1.5 | 1 | 9 |
| | *Arctosa alpigena lamperti* | 1 | 3 | 1 | 1 | 10 |
| Pisauridae | *Dolomedes tenebrosus* | 1 | 1 | 1 | 2 | 13 |
| Zoropsidae | *Tengella perfuga* | 0 | 4.5 | 3 | 2.5 | 12 |
| Dictynidae | *Argyroneta aquatica* | 0 | 4 | 2 | 1 | 6 |
| Agelenidae | *Eratigena atrica* | 0 | 4.5 | 2 | 1 | 9 |
| Phyxelididae | *Phyxelida tanganensis* | 0 | 4 | 3 | 4 | 8 |
| Uloboridae | *Hyptiotes paradoxus* | 0 | 5 | 3 | 4 | 6 |
| Tetragnathidae | *Metellina segmentata* | 0 | 5 | 4 | 3 | 5 |
| Mimetidae | *Mimetus puritanus* | 1 | 2.5 | 1 | 1.5 | 7 |
| | *Mimetus notius* | 1 | 2.5 | 1 | 1.5 | 7 |
| Araneidae | *Neoscona theisi* | 0 | 5 | 4 | 3 | 7 |
| | *Araneus cavaticus* | 0 | 5 | 4 | 3 | 12 |
| | *Araneus diadematus* | 0 | 5 | 4 | 3 | 10 |
| | *Larinioides cornutus* | 0 | 5 | 4 | 3 | 7 |
| Theridiidae | *Enoplognatha ovata* | 0 | 5.5 | 4 | 3 | 4 |

similar to sit & wait, is defined as an intermediate state between sit & wait and active hunting (*Schmitz & Suttle, 2001*; *Miller, Ament & Schmitz, 2014*). With a sit & wait strategy, the otherwise motionless spider grabs the prey when it comes within striking distance, whereas with sit & pursue, the otherwise motionless spider runs after the prey when it comes within several centimeters proximity and pursues the prey until captured (*Schmitz & Suttle, 2001*; *Miller, Ament & Schmitz, 2014*).

3. Silk: Main type of silk used, scored as 1: none/MAP dragline, 1.5: burrow, 2: aciniform sheet, 3: cribellate, or 4: viscous silk, i.e., that produced from the aggregate gland spigots.

4. Type: A measure of the variety of spigots the species possessed beyond what we called the standard 7. These scored as 1: standard 7, 1.5: standard 7 + modified piriform gland spigots on the ALS, 2: standard 7 + modified spigot on PLS, 2.5: standard 7 + modified spigot on PLS with two 'flankers' + cribellum, 3: standard 7 + aggregate and

flagelliform gland spigots on the PLS or 4: standard 7 + cribellum, paracribellar gland spigots on the PMS and a pseudoflagelliform gland spigot on the PLS.

5. Instar: Maximum number of instars that a species goes through to reach adulthood (females).

The twelve dependent variables were continuous and were the average number of spigots for each of the 'standard 7' found in all the spider species spinning fields. We focused on the adult female instar as well as the second instar, i.e., when all spiders emerge from the egg sac. Specifically, for adult females, the seven dependent variables were average number of spigots for: 1. MAP gland spigots on the ALS, 2. PI gland spigots on the ALS, 3. mAP gland spigots on the PMS, 4. AC gland spigots on the PMS, 5. CY gland spigots on the PMS, 6. AC gland spigots on the PLS and 7. CY gland spigots on the PLS. For second instars, the five dependent variables were the average number of spigots for: 8. MAP gland spigots on the ALS, 9. PI gland spigots on the ALS, 10. mAP gland spigots on the PMS, 11. AC gland spigots on the PMS and 12. AC gland spigots on the PLS. Cylindrical gland spigots are only found in adult female spiders, which is why they are not included with the second instars.

Phylogenetic comparative analyses have typically dealt with at least two continuous variables. Methods have improved to accommodate the increase in Type 1 Error associated with discrete variables such that analyses like phylogenetic generalized least square models (PGLS) can be robustly performed (*Graber, 2013*; *Maddison & FitzJohn, 2015*). While theoretically repeated-measures or a factorial ANOVA are possible in a phylogenetic context, the methods of incorporating them into phylogenetic comparative analyses are not yet developed (*Guo et al., 2007*). Thus, our time series study could not be analyzed as a whole unit. To gain a picture of the potential effects of ontogeny, we performed several PGLS analyses on the adult female spigot numbers as well as those of the second instars. These include the stage at which all spiders emerge from the egg sac and are the most similar in condition (second instars) to the stage when the most diversification and growth in spigots appears (adult female stage).

### Phylogeny

We used the topology of the recently published Araneae Tree of Life (AToL) (*Wheeler et al., 2016*) for the 22 species included in our study. At the time of publication of this manuscript, the AToL sequences have not been publicly released. Thus, while we were not able to generate a time-calibrated tree or one with branch lengths equal to a rate of molecular evolution, we could use the published topology and create two phylograms. One phylogram had all branch lengths equal to 1 and the other was an ultrametric tree with the same topology (Fig. 2), using the *ape* (version 4.1) package in R (*Paradis, Claude & Strimmer, 2004*). Both trees were used in model testing for phylogenetic comparative analyses and ultimately, we used the ultrametric topology in our final analyses.

### Phylogenetic comparative analyses

All analyses were performed in *R* using *RStudio* (version 3.2, *R Core Team, 2016*; *RStudio, 2017*). First, we tested both sets of dependent and independent variables for phylogenetic signal using the *phytools* (version 0.5–64) package in R (*Revell, 2012*). All independent

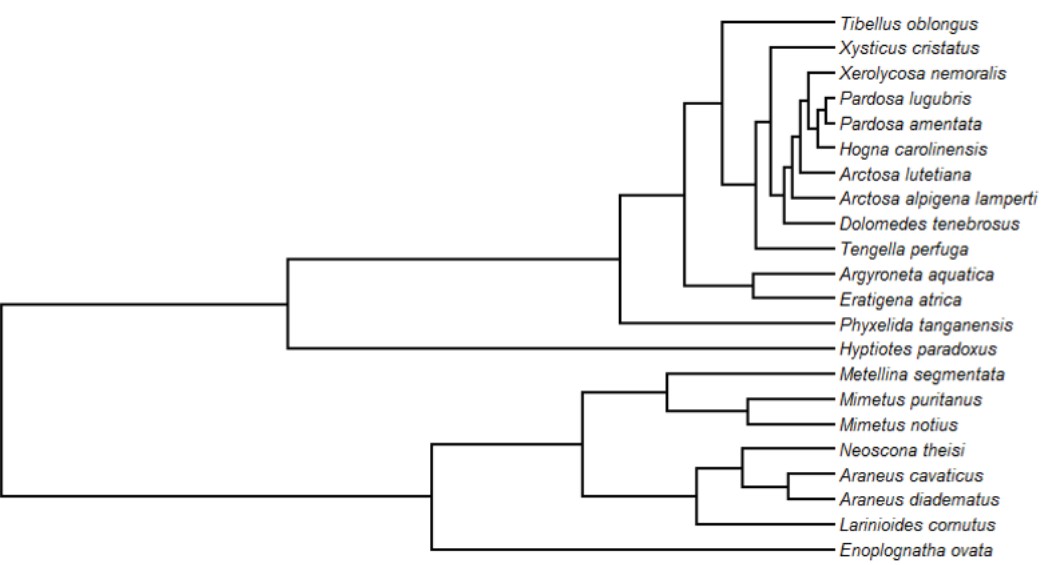

**Figure 2 Phylogenetic tree used in PGLS analyses.** Ultrametric tree with 22 species, topology follows the AToL (*Wheeler et al., 2016*).

variables had strong phylogenetic signal with *Blomberg's K* (*Blomberg, GarlandJr & Ives, 2003*) and *Pagel's* λ (*Pagel, 1999*) being close to 1 and with significant *p*-values while most of the dependent variables did not show phylogenetic signal (Table 3). Despite this, given the strong signal in the predictor variables, we decided to proceed with PGLS analyses. Because the *K* values were so close to 1, a Brownian motion model of evolution was the best fit. We further tested various models of evolutionary rate, and confirmed the Brownian model of evolution being the best fit (Results S1). Next, using both trees, we tested PGLS models using generalized least squares method of model selection with single term up to all five independent variables included. We used the *R* packages, *nlme* and *MuMIn* (version 1.15.6) to test model fit (*Bartón, 2016*; *Pinheiro et al., 2017*). PGLS analyses required the use of the *ape*, *geiger* (version 2.0.6), *nlme* (version 3.1–131) and *phytools* packages in *R* (*Paradis, Claude & Strimmer, 2004*; *Harmon et al., 2008*; *Revell, 2012*; *Pinheiro et al., 2017*). With delta AICc values equal to zero or very close to zero between terms, and a significant *p*-value associated with the single term Instar, we determined that single term models were the best fit with the trees and the datasets for PGLS as there was no significant effect of adding additional terms (Table 4, Results S2). We ran a total of 35 PGLS analyses covering each of the adult female dependent variables (average numbers of specific silk spigots) and 25 PGLS analyses covering each of the second instar dependent variables using the *ape* and *geiger* packages in *R* (*Paradis, Claude & Strimmer, 2004*; *Harmon et al., 2008*). We did this with both trees, and while the specific analyses results differed, the main conclusions did not. We also performed ANOVA analyses on the independent variable means derived through the PGLS models. Because all the independent variables were discrete the ANOVA analyses served to corroborate the PGLS results. Here we report the significant PGLS coefficient of correlation results (Table 5) using a maximum likelihood approach, with

**Table 3 Results of tests for phylogenetic signal.** Results of the tests for phylogenetic signal using Pagel's λ and Blomberg's K for all independent and dependent variables. Those with significant phylogenetic signal are bolded and include all independent variables and three of the response variables.

| Variable | Pagel's λ | *P*-value | Blomberg's K | *P*-value |
|---|---|---|---|---|
| **(I) Strategy** | **1.000** | **0.0006** | **0.955** | **0.001** |
| **(I) Specific** | **1.000** | **0.002** | **0.904** | **0.002** |
| **(I) Silk** | **1.000** | **0.0003** | **0.915** | **0.001** |
| **(I) Type** | **1.000** | **0.00008** | **1.243** | **0.001** |
| **(I) Instar** | 0.565 | 0.139 | **0.512** | **0.012** |
| **(D) Fem ALS MAP** | **1.000** | **0.00004** | **0.634** | **0.009** |
| (D) Fem ALS PI | 0.110 | 0.698 | 0.158 | 0.795 |
| **(D) Fem PMS mAP** | **1.000** | **5.36 e $^{-10}$** | **3.834** | **0.002** |
| (D) Fem PMS AC | 0.532 | 0.210 | 0.176 | 0.769 |
| (D) Fem PMS CY | 0.229 | 0.141 | 0.452 | 0.076 |
| (D) Fem PLS AC | 0.572 | 0.270 | 0.184 | 0.731 |
| (D) Fem PLS CY | 0.057 | 0.698 | 0.315 | 0.243 |
| (D) 2nd ALS MAP | 6.61E−05 | 1.000 | 0.313 | 0.287 |
| (D) 2nd ALS PI | 0.076 | 0.603 | 0.093 | 0.877 |
| **(D) 2nd PMS mAP** | **1.000** | **0.002** | **0.794** | **0.008** |
| (D) 2nd PMS AC | 6.61E−05 | 1.000 | 0.084 | 0.871 |
| (D) 2nd PLS AC | 6.61E−05 | 1.000 | 0.081 | 0.895 |

a Brownian model of evolution and the ultrametric tree. The 60 full PGLS analyses and ANOVA results using the ultrametric tree are available in Results S3 and S4.

## Ancestral character estimation

Finally, we used the ultrametric tree and the *ace* function in the *ape* package in *R* (*Paradis, Claude & Strimmer, 2004*) to conduct ancestral character estimation on the diverse, singular spigots found on the PLS to explore the unresolved question of whether these spigots are homologous structures or not. The spigots of interest were the modified spigot (MS) found in some cribellate and ecribellate spiders, the pseudoflagelliform gland spigot (PF) found in cribellate orb weavers and the flagelliform gland spigot (FL) found in viscous orb weavers. We added five additional taxa, using data derived from adult female SEM images in *Griswold et al. (2005)* to allow for broader taxon sampling deeper into the phylogeny. These species were the cribellate *Hypochilus pococki* N.I. Platnick (1987; Hypochilidae), *Kukulcania hibernalis* (N.M. Hentz, 1842; Filistatidae), *Thaida peculiaris* F. Karsch (1880; Austrochilidae), *Megadictyna thilenii* F. Dahl (1906; Megadictynidae) and ecribellate *Nicodamus mainae* M.S. Harvey (1995; Nicodamidae). The Nicodamoidea (Megadictynidae plus Nicodamidae) are sister to all Araneoidea, while the other families are sister to the Araneoidea + RTA clades (*Wheeler et al., 2016*). We used a maximum likelihood method with a model of the weighted rate matrix of substitutions for these spigots (Table 6). As we describe below, we constrained our rate matrix (Table 6) that we used to model substitution rates across branch lengths based on prior knowledge about historical possession of the flagelliform gland, pseudoflagelliform gland and modified

**Table 4 Significant results of Model Selection for PGLS analyses.** Model selection results for selected single-term PGLS models, showing Instar as most important and the model selection as significant. $AIC_c$ and $\Delta AIC_c$ values for model selection for all independent variables, by response variable. Term codes are: Instar = 1, Type = 2, Silk = 3, Specific = 4, Strategy = 5. Significant results are bolded, if not significant, but most important term, they are bold italicized.

| Term | $AIC_C$ | $\Delta$ | Weight |
|---|---|---|---|
| **Female ALS Piriform Model Selection:** | | | |
| *1* | **238.30** | **0.00** | **1.00** |
| 2 | 255.39 | 17.09 | 0.00 |
| 5 | 258.27 | 19.98 | 0.00 |
| 3 | 259.17 | 20.88 | 0.00 |
| 4 | 260.27 | 21.97 | 0.00 |
| **2nd Instar ALS MAP Model Selection:** | | | |
| *Instar* | **43.35** | **0.00** | **0.84** |
| *Strategy* | 48.75 | 5.40 | 0.06 |
| *Type* | 49.32 | 5.97 | 0.04 |
| *Specific* | 49.35 | 5.99 | 0.04 |
| *Silk* | 50.57 | 7.21 | 0.02 |
| **Female PLS Aciniform Selection:** | | | |
| *1* | **251.95** | **0.00** | **0.91** |
| 4 | 259.03 | 7.08 | 0.03 |
| 5 | 259.18 | 7.23 | 0.02 |
| 3 | 259.58 | 7.62 | 0.02 |
| 2 | 259.83 | 7.88 | 0.02 |
| **2nd Instar ALS Piriform Model Selection:** | | | |
| *1* | *205.31* | *0.00* | *0.37* |
| 3 | 206.80 | 1.49 | 0.18 |
| 5 | 207.03 | 1.72 | 0.16 |
| 4 | 207.13 | 1.82 | 0.15 |
| 2 | 207.15 | 1.83 | 0.15 |
| **Female PMS Aciniform Model Selection:** | | | |
| *1* | **258.04** | **0.00** | **0.98** |
| 2 | 268.36 | 10.32 | 0.01 |
| 5 | 268.51 | 10.47 | 0.01 |
| 3 | 268.70 | 10.66 | 0.00 |
| 4 | 268.74 | 10.70 | 0.00 |

spigots in each taxon (Table 6) (*Platnick & Griswold, 1991*; *Griswold et al., 2005*; *Dimitrov et al., 2016*; *Wheeler et al., 2016*). Some of the more recently derived clades within the Araneoids (*Dimitrov et al., 2016*) have lost flagelliform gland spigots (i.e., Mimetidae, Arkyidae). We ranked the transition to Loss (of spigots) as 1 substitution and that to none, pseudoflagelliform, or modified spigots (not observed anywhere in this lineage) as 0 substitutions, or no likelihood (Table 6). Because they are found in cribellate groups ancestral to both the RTA and Araneoid clades, we ranked all transitions for modified spigots to the other four states as 1 substitution (Table 6). One interpretation of Dollo's

**Table 5 Significant results of PGLS analyses.** Significant results of the phylogenetic generalized least squares analyses testing for correlation of independent variables with the twelve dependent variables (average number of each standard spigot) for both adult female datasets and second instar datasets. The coefficient values provide the correlation coefficient of the means of the independent variable with the dependent variable.

| | | | |
|---|---|---|---|
| **Second Instar ALS MAP** | | | |
| *Model: Average ~ Instar* | | | |
| **PGLS Coefficient:** | | *t* | *P* |
| Instar | 0.095 | 2.283 | 0.034 |
| **Second Instar PMS mAP** | | | |
| *Model: Average ~ Instar* | | | |
| **PGLS Coefficient:** | | *t* | *P* |
| Instar | 0.104 | 2.641 | 0.016 |
| **Female PMS mAP** | | | |
| *Model: Average ~ Strategy* | | | |
| **PGLS Coefficient:** | | *t* | *P* |
| Strategy | 0.291 | 2.448 | 0.024 |
| **Female ALS Piriform** | | | |
| *Model: Average ~ Instar* | | | |
| **PGLS Coefficient:** | | *t* | *P* |
| Instar | 17.204 | 5.355 | <0.000 |
| *Model: Average ~ Type* | | | |
| **PGLS Coefficient:** | | *t* | *P* |
| Type | 61.692 | 4.413 | 0.0003 |
| **Female PMS Aciniform** | | | |
| *Model: Average ~ Instar* | | | |
| **PGLS Coefficient:** | | *t* | *P* |
| Instar | 11.725 | 2.857 | 0.010 |

**Table 6 Evolutionary rate matrix for ACE of unique PLS spigots.** The substitution rate matrix for the spigots of the PLS used as the model for the ancestral character estimation analysis. The rows are the from direction, while the columns are the to direction for state changes.

| SPIGOT | Flagelliform | Loss | Modified | None | Pseudoflagelliform |
|---|---|---|---|---|---|
| Flagelliform | – | 1 | 0 | 0 | 0 |
| Loss | 0 | – | 0 | 0 | 0 |
| Modified | 1 | 1 | – | 1 | 1 |
| None | 1 | 0 | 1 | – | 1 |
| Pseudoflagelliform | 1 | 1 | 0 | 0 | – |

Law is that it is easier to lose a structure than to re-evolve it (*Dollo, 1893*) and this influenced how we weighted the remaining matrix for Pseudoflagelliform, Loss, and the possible None (no spigot) cases. Since one of the cribellate ancestral species, *Kukulcania hibernalis,* does not possess anything corresponding to a modified or pseudoflagelliform gland spigot, and is classified as 'None', we conservatively allowed for a single substitution from None to Flagelliform, Modified and Pseudoflagelliform in our rate matrix (Table 6). Finally, since

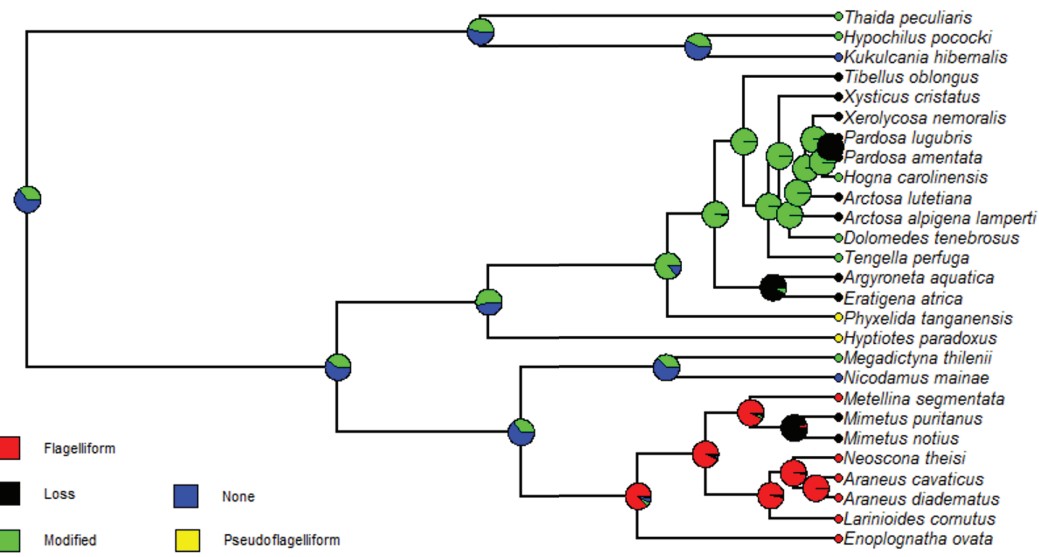

**Figure 3** **Ancestral character estimation results for the unique PLS spigots.** Phylogram with ancestral character estimation of singular PLS spigots on the ultrametric tree with five additional taxa. The root node is estimated to either possess a modified spigot (green) or no specialized spigots (blue).

the cribellate orb weaving sister group to the RTA clade possesses a pseudoflagelliform gland spigot (PF) and RTA clade members possess modified (MS) or no spigots, we allowed for 0 substitutions in this direction. We also allowed for a substitution rate of 1 between Pseudoflagelliform and Flagelliform as we do not know whether the ancestral orb weaver possessed pseudoflagelliform or modified spigots (*Bond et al., 2014*; *Garrison et al., 2016*; *Wheeler et al., 2016*). We then plotted the likelihoods of states at each node on the ultrametric phylogeny (Fig. 3, Results S5).

Additionally, we performed ancestral character estimation analyses for four of the five independent variables from the PGLS analyses on the ultrametric tree with increased taxa: Strategy, Specific, Silk and Type (Results S5). The constrained rate matrices for each variable along with explanations for the given matrices are provided in Table S2. We did not include maximum instar, as those data were not available in the literature for the five additional taxa. Also, due to its variability within a species (see Table S1), ancestral character estimation may not be a useful tool to use to understand maximum instar in silk evolution.

# RESULTS

## Spigot ontogeny of *Dolomedes tenebrosus* and *Hogna carolinensis*

All instars of *D. tenebrosus* (instars 2–13) and more than half of the instars (instars 2–7 and 12) of *H. carolinensis* were observed and sampled from lab colonies for SEM imaging to assess the spigot ontogeny of the full spinning field of these lycosoid spiders (*Homann, 1971*; *Griswold, 1993*; *Polotow, Carmichael & Griswold, 2015*). *Dolomedes tenebrosus* reached adulthood in thirteen instars while *H. carolinensis* reached adulthood in twelve (Table 1).

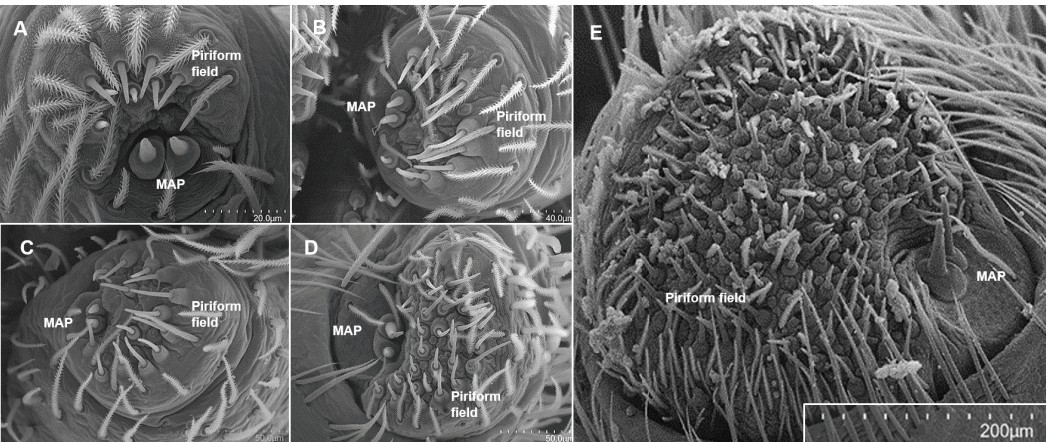

**Figure 4** **SEMs of the anterior lateral spinnerets.** ALS spinning field of selected instars of *Dolomedes tenebrosus* and Hogna *carolinensis*: (A) *D. tenebrosus* 3rd instar left ALS; (B) *H. carolinensis* 4th instar (left ALS); (C) *H. carolinensis* 5th instar (left ALS); (D) *D. tenebrosus* 8th instar (left ALS); (E) *D. tenebrosus* 12th instar (right ALS, penultimate female). MAP, Major ampullate gland spigot.

### Anterior lateral spinnerets

Both species possessed two MAP gland spigots, except for the adult male stage (Table 1, Fig. 4). Piriform gland spigots increased in number to adulthood (Fig. 4). However, in adult male *D. tenebrosus,* PI gland spigots decreased from the penultimate instar, which led to sexual dimorphism (Table 1). It was not until later instars of *D. tenebrosus* and *H. carolinensis* that the number of PI gland spigots increased in greater magnitude from instar to instar (Figs. 4A–4E, Table 1). We also observed sensilla (sensory pores) in the MAP fields of both species (Fig. 5).

### Posterior median spinnerets

Both species possessed two mAP gland spigots, except for the adult male stage of *D. tenebrosus* (Table 1, Fig. 6). CY gland spigots did not appear until the adult female and probably penultimate female instars. Penultimate *D. tenebrosus* possessed at least one pre-cylindrical gland spigot (Fig. 6F). *D. tenebrosus* adult females bore many more CY gland spigots than *H. carolinensis* (Fig. 6C). In *D. tenebrosus*, AC gland spigot numbers did not change at 4 AC gland spigots for instars 2/3, then at 5 AC gland spigots through instars 4/5/6, and then 8 AC gland spigots for instars 8/9/10 (Table 1, Figs. 6D, 6E). Both males and females lost AC gland spigots in the final molt to adulthood. In *H. carolinensis*, AC gland spigots dropped in number at instar 4 and remained at 3 AC gland spigots for the next two instars (Table 1, Figs. 6A, 6B). By the adult instar, the number was far greater in the female *H. carolinensis* than the female *D. tenebrosus* (Table 1, Fig. 6C).

### Posterior lateral spinnerets

Cylindrical gland spigots also appeared in the penultimate female stage with the three pre-spigots visible (Fig. 7F). Adult female *H. carolinensis* possessed a single CY gland spigot compared to the 28 CY gland spigots in adult female *D. tenebrosus* (Table 1). Aciniform

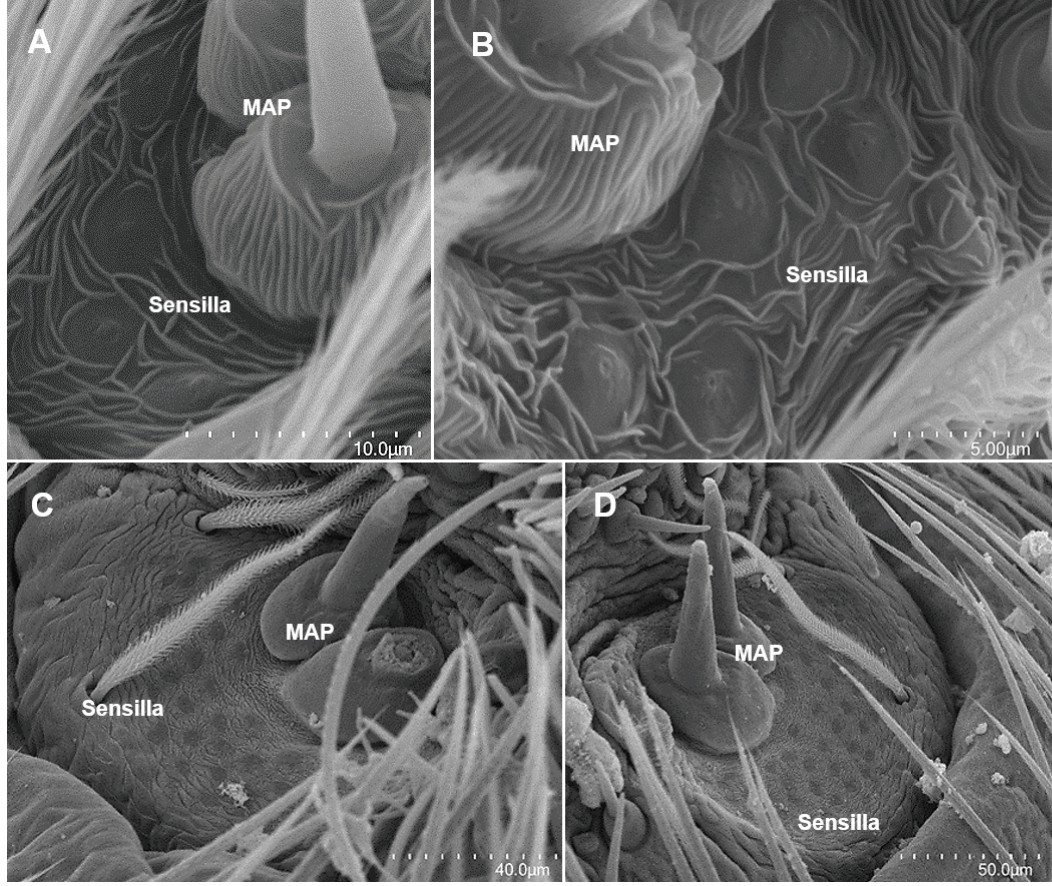

**Figure 5** **SEMs of the sensilla of the anterior lateral spinnerets.** Pore field of the MAP on the ALS of selected instars of *Dolomedes tenebrosus* and *Hogna carolinensis*: (A) *H. carolinensis* 3rd instar (left ALS); (B) *H. carolinensis* 5th instar (left ALS); (C) *D. tenebrosus* 11th instar (penultimate male, left ALS); (D) *D. tenebrosus* 12th instar (penultimate female, right ALS). MAP, Major ampullate gland spigot; Sensilla, sensory pores in MAP field.

gland spigots presented different trends between the two species (Table 1, Figs. 7 and 8). In *D. tenebrosus*, AC gland spigot numbers slowly increased with the same number persisting for 2–3 instars, then increasing (Fig. 7). We also observed loss of spigots in the final molts to adulthood in both males and females (Table 1). In *H. carolinensis*, a sharp decrease to 3 AC gland spigots occurred in instar 4 and persisted through instar 5, then increased to 7 AC gland spigots in instar 6 persisting through instar 7 (Table 1, Fig. 8). The female *H. carolinensis* possessed more AC gland spigots than the female *D. tenebrosus*. In both species, a larger spigot was tentatively identified as a 'modified spigot' (see *Griswold et al., 2005*: 61; character 96), with a potential pre-modified spigot observed in the penultimate female stage of *D. tenebrosus*. These made no other appearance in the ontogeny the spinning fields of either species (Table 1, Figs. 7 and 8).

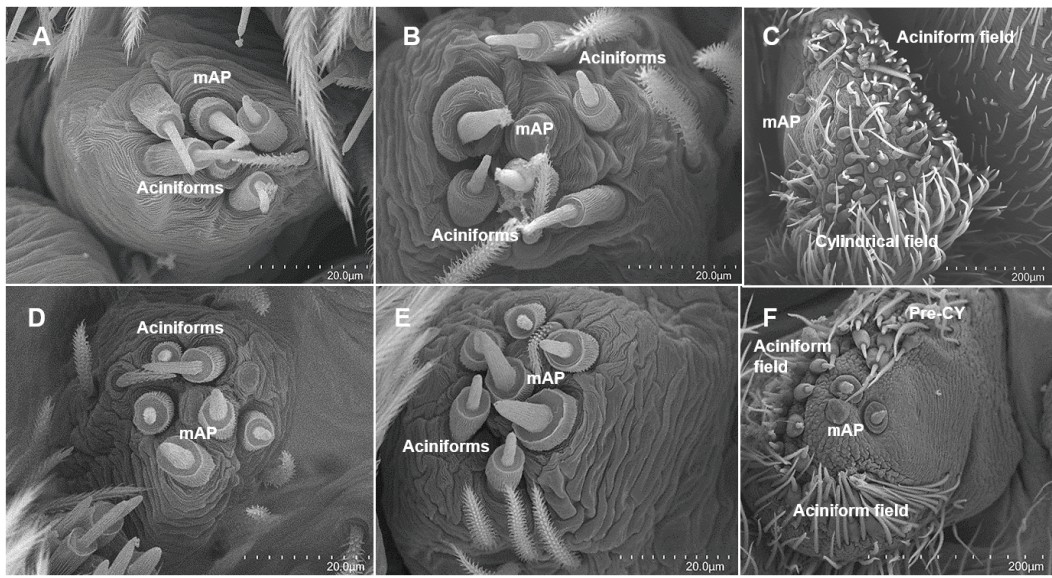

**Figure 6** **SEMs of the posterior median spinnerets.** PMS spinning field of selected instars of *Dolomedes tenebrosus* and *Hogna carolinensis*: (A) *H. carolinensis* 2nd instar (left PMS); (B) *H. carolinensis* 6th instar (right PMS); (C) *H. carolinensis* 12th instar, female (left PMS); (D) *D. tenebrosus* 3rd instar (left PMS); (E) *D. tenebrosus* 5th instar (right PMS); (F) *D. tenebrosus* 12th instar, penultimate female (right PMS). mAP, Minor ampullate gland spigot; Pre-CY, pre-cylindrical gland spigot.

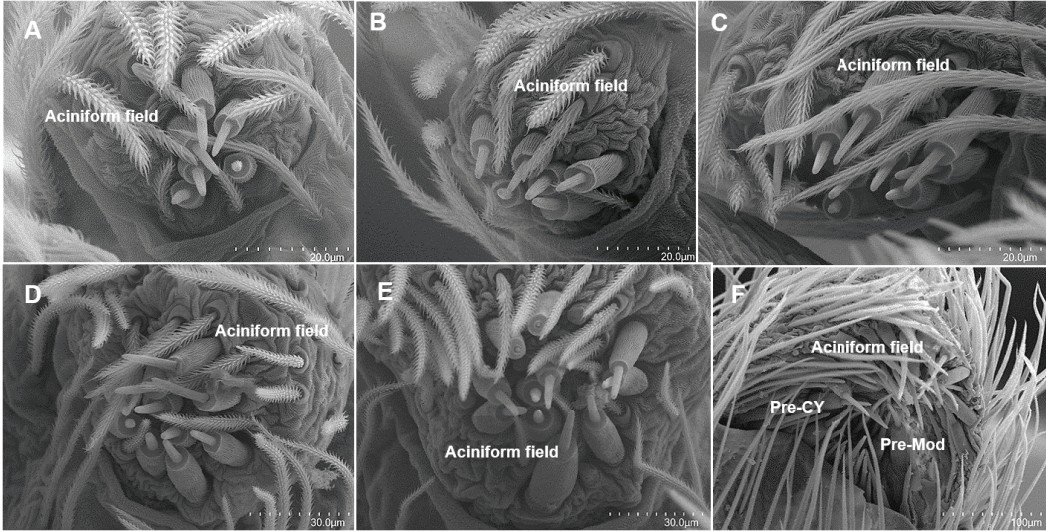

**Figure 7** **SEMs of the posterior lateral spinneret spigots of *Dolomedes tenebrosus*.** PLS spinning field of selected instars of *Dolomedes tenebrosus*, early instars have low conserved numbers of aciniform gland spigots, which suddenly increase at the penultimate instar: (A) 3rd instar (left PLS); (B) 4th instar (right PLS); (C) 6th instar (left PLS); (D) 8th instar (left PLS); (E) 10th instar, antepenultimate female (left PLS); (F) 12th instar, penultimate female (left PLS). Pre-CY, Pre-cylindrical gland spigots; Pre-Mod, Pre-Modified spigot.

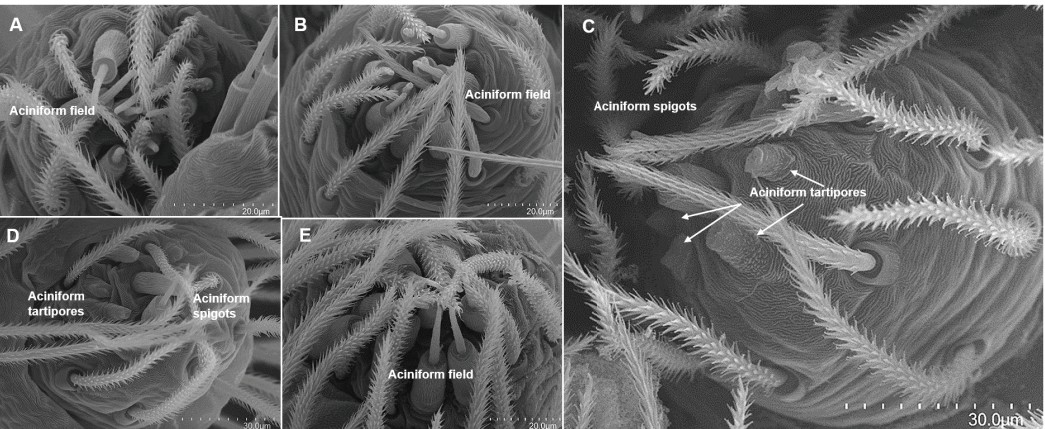

**Figure 8** **SEMs of the posterior lateral spinneret spigots of *Hogna carolinensis*.** Right PLS spinning field of selected instars of *Hogna carolinensis*, early instars have higher numbers of aciniform gland spigots, lose them over two molts, then begin to gain them back again: (A) 2nd instar; (B) 3rd instar; (C) 4th instar; (D) 5th instar; (E) 6th instar. Aciniform tartipores = cuticular scars from aciniform gland spigots present in the previous instar.

## Phylogenetic comparative analyses

We compiled a full spigot ontogeny dataset of 22 species representing thirteen spider families and scored the five independent, predictor variables: Strategy, Specific, Silk, Type, Instar (Table S1, Table 2). After determining that the independent variables had strong phylogenetic signal, whereas only three of the twelve dependent response variables did, we decided to proceed with the phylogenetic generalized least squares analyses (Table 3). After thorough tree and model selection analyses, we determined that single term models were the best fit along with the ultrametric tree and a Brownian model of evolution (Table 4, Fig. 2). We ran a total of 60 PGLS analyses (Results S3, S4) and here report the significant results of those analyses (Table 5).

In the adult female analyses, we found that Instar was a significant predictor in a few cases. For piriform gland spigots (ALS), the coefficient of correlation between Instar and average number of spigots was significant (Table 5). This was also the case for aciniform gland spigots of the PMS (Table 5). For female PI gland spigots, Type, or the variety of spigots possessed, was also a significant predictor of Average number of PI (Table 5). Of interest, the female mAP gland (PMS) spigots did show strong phylogenetic signal and Strategy was a significant predictor of Average number of mAP gland spigots. This means that, after the bias of the correlation due to phylogeny was accounted for through the PGLS, the species classification of either being a web builder or not had a strong coefficient of correlation with the average number of mAP gland spigots possessed (Table 5).

In the second instar analyses, we found that Instar was also a significant predictor or independent variable. For MAP gland (ALS) spigots in second instars, the coefficient of correlation between Instar and Average number of MAP gland spigots was significant (Table 5). Spiders possessed one, two or no MAP gland spigots at the emergence from the egg sac (second instar) (Table S1). For the mAP gland (PMS) spigots, which did show

strong phylogenetic signal in preliminary testing, Instar was a significant predictor of the Average number of mAP gland spigots (Table 5). Second instar spiders possessed two, one or no mAP gland spigots at this first stage outside of the egg sac (Table S1).

## Ancestral character estimation of singular spigots on the PLS

We performed a maximum likelihood ancestral character estimation (ACE) of potentially homologous spigots producing axial lines, i.e., the singular spigot on the PLS. We used the ultrametric tree and overall scoring of whether a species possesses a spigot and if so, which spigot it is: Flagelliform, Modified, None (never historically possessing a singular spigot), Pseudoflagelliform, or Loss (previously possessing a singular spigot) of a spigot. We used a constrained rate matrix for the five states as our model of substitution rates for the ACE analyses (Table 6). The log likelihood value for the analysis was −46.023. Within this analysis we calculated the scaled likelihoods of states at the root (Flagelliform gland spigot: 0.000, Modified spigot: 0.365, None (no singular spigot that produces axial lines): 0.635, Pseudoflagelliform gland spigot: 0.000, Loss: 0.000), as well as the other nodes of the phylogram produced (Fig. 3, Results S5). In our color-coded results, the predominately red clade was the Araneoidea, which includes viscous orb weavers (FL spigot) and the pirate spiders (Mimetidae) which do not possess a FL spigot as adults (Loss) (Fig. 3). However, as very young juveniles mimetids possessed vestigial FL and aggregate gland spigots on the PLS (Table S1, *Townley & Tillinghast, 2009*). The predominately green clade (MS spigot) was the RTA clade, with loss of MS spigot (black) in many of the tip species, while just sister to that were the yellow clades (Pseudoflagelliform), which included the cribellate sheet (*Phyxelida*) and cribellate orb weavers (*Hyptiotes*). Additional plesiomorphic cribellate taxa, possessing MS spigots (green, *Thaida*, *Hypochilus*) or no unique spigot on the PLS (blue, *Kukulkania*) were included for deeper phylogenetic sampling (Fig. 3). Finally, the results suggested that, at the ancestral root node, the PLS singular spigot was more likely to have been a modified spigot than a pseudoflagelliform gland spigot, and that the ancestor either possessed a modified spigot or no singular spigot (Fig. 3, Table 4).

We followed a similar procedure for maximum likelihood ancestral character state estimation for four of the five independent variables: Strategy, Specific, Silk and Type. The constrained rate matrices used as the model for the analyses are given in Table S2. The resulting state likelihood values for each node are also given in Results S5 with the previous ACE node results. For the variable Strategy (foraging strategy), the ancestral character estimation model has a log likelihood of −14.73. At the root, the ancestral state was "Web" or web building (likelihood = 1.0) rather than no web or loss of web (Fig. 9, Results S5). Therefore, the ancestral state for the araneoids and lycosoids was a web builder. Specific (specific foraging strategy) ACE model has a log likelihood of −58.65. At the ancestral node, the states are for web building (Fig. 10). The ancestor of lycosoids and araneoids likely spun a funnel web (likelihood = 0.812) or a sheet web (likelihood = 0.084); an orb web (likelihood = 0.105) is less likely (Fig. 10, Results S5). The ACE model for Silk (main type of silk expressed) has a log likelihood of −44.61. The ancestor of araneoids and lycosoids produced cribellate silk (likelihood = 1.0; Fig. 11, Results S5). Finally, the ACE model for Type (variety of silk spigot types) had a log likelihood of −36.14. The ancestral
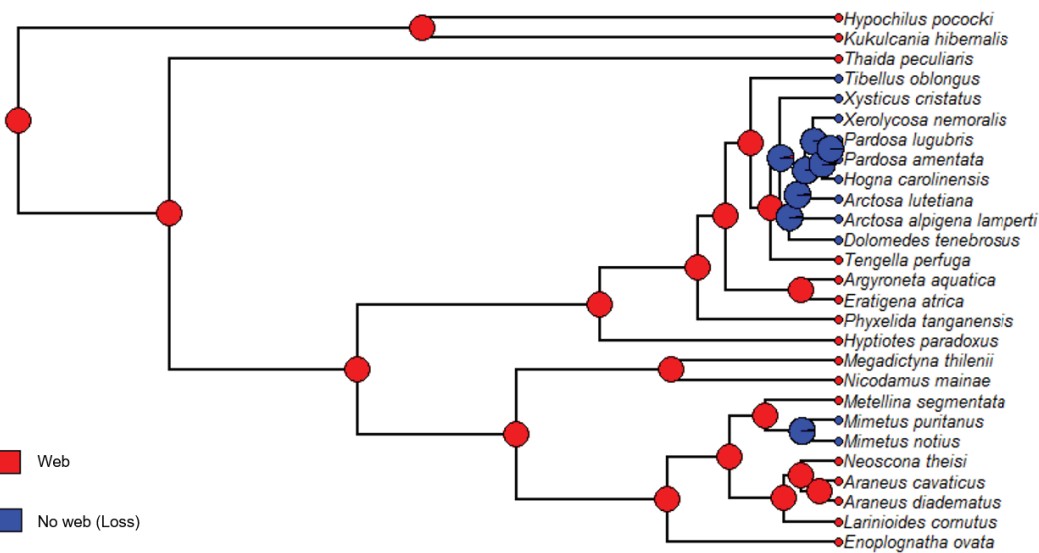

**Figure 9  ACE results for foraging strategy.** Phylogram with ancestral character estimation of foraging strategy (Strategy), web or no web (loss) on the ultrametric tree with five additional taxa. Clearly, the ancestral state was web spinning (red).

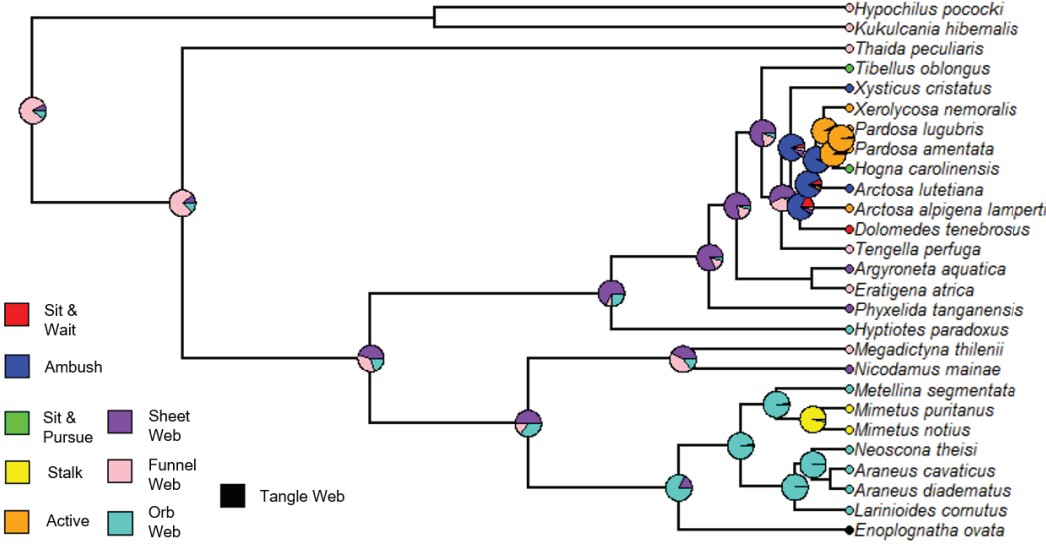

**Figure 10  ACE results for specific foraging strategies.** Phylogram with ancestral character estimation of specific foraging strategies on the ultrametric tree with five additional taxa. Specific strategies include: sit & wait, ambush, sit & pursue, stalking, active, sheet web, funnel web, orb web and tangle web.

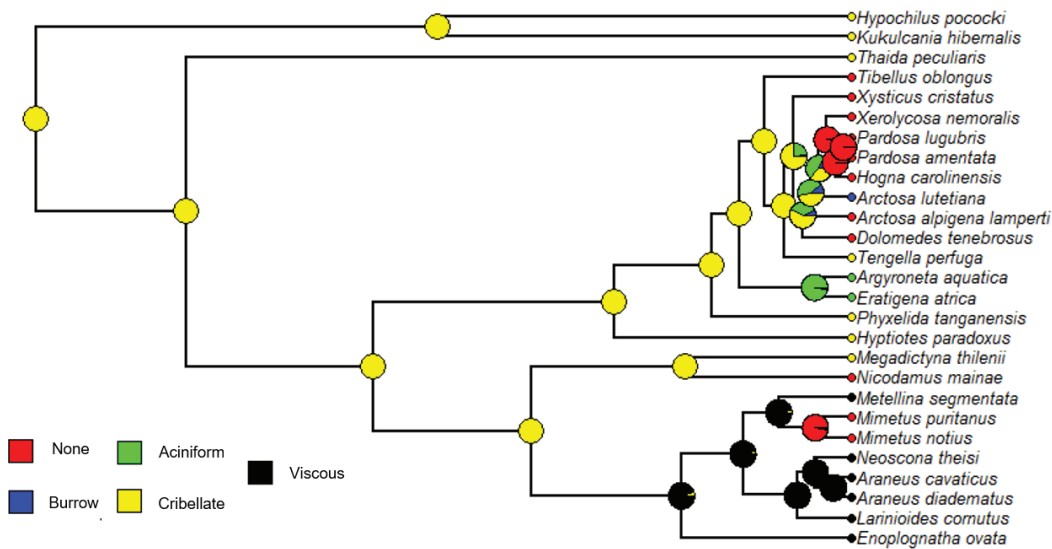

**Figure 11** **ACE results for main type of silk utilized by a species.** Phylogram with ancestral character estimation of Silk (main type of silk expressed) on the ultrametric tree with five additional taxa. Silk types include: none, burrow, aciniform, cribellate and viscous. This analysis suggests that the ancestral state for these clades is cribellate (yellow).

state at the root node was undetermined, but the two states with the highest likelihoods were cribellate spiders possessing either Standard 7 + Cribellum + Paracribellar spigots (likelihood = 0.398) or Standard 7 + Cribellum + Pseudoflagelliform + Paracribellar spigots (likelihood = 0.433). All additional five states had much lower likelihoods (Fig. 12, Results S5).

# DISCUSSION

This is the first published full ontogeny of the spinning apparatus of both *D. tenebrosus* and *H. carolinensis*. This is also the first statistical phylogenetic comparative analysis exploring questions in silk use and evolution across several spider taxa. By creating a standardized dataset across 22 species, we could unite the few existing spigot ontogeny studies into a comparative and phylogenetic context (*Wąsowska, 1977*; *Yu & Coddington, 1990*; *Hajer, 1991*; *Townley & Tillinghast, 2009*; *Dolejš et al., 2014*; R Carlson & CE Griswold, 1996, unpublished data; RE Alfaro, 2017, unpublished data).

## Spigot ontogeny of *D. tenebrosus* and *H. carolinensis*

Both *D. tenebrosus* and *H. carolinensis* are large-bodied lycosoids. *Dolomedes tenebrosus* belongs to the Pisauridae family, the fishing or nursery web spiders, and employs a sit & wait foraging strategy (Table 2) whereas *H. carolinensis*, belonging to the diverse Lycosidae family, employs a sit & pursue strategy (Table 2) (*Schmitz & Suttle, 2001*; *Miller, Ament & Schmitz, 2014*). While both have a similar number of instars to adulthood, the two species differ dramatically from each other in the loss and regain of aciniform gland spigots on the PMS and PLS (Table 1, Figs. 6, 7 and 8). In *D. tenebrosus*, AC gland spigot numbers on the

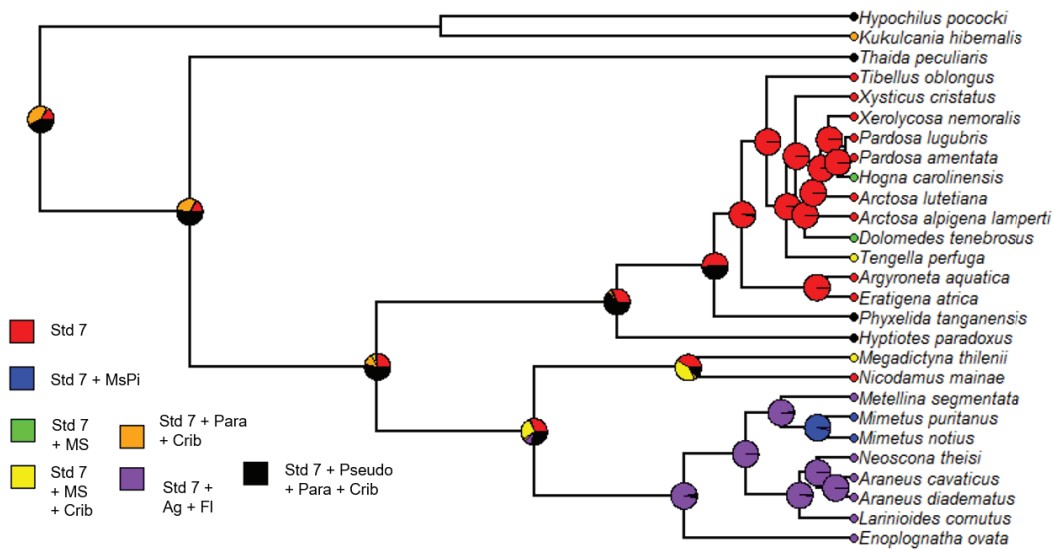

**Figure 12 ACE results for variety of spigots possessed by a species.** Phylogram with ancestral character estimation of Type of spigots (variety of spigot types possessed) on the ultrametric tree with five additional taxa. These included several states, including the standard seven spigots, along with additional spigots increasing in diversity. Std 7, standard 7; MsPi, modified piriform gland spigots; MS, modified spigot; Crib, cribellum; Para, paracribellar spigots; Ag, aggregate gland spigots; Fl, flagelliform gland spigot; Pseudo, pseudoflagelliform gland spigot.

PMS do not change for multiple instars slowly increasing after two to three instars with a dramatic increase in the final molts. In *H. carolinensis,* AC gland spigot numbers drop and remain low for two to three instars before gradually increasing, until a similar dramatic increase in number with the final molt (Table 1, Fig. 6). The same interesting trend was also observed for aciniform gland spigots on the PLS in both species (Table 1, Figs. 7 and 8). These trends were not observed in other lycosoids previously studied including other members of the Lycosidae (*Wąsowska, 1977*; *Dolejš et al., 2014*). *H. carolinensis* do not form webs. *D. tenebrosus* instars were observed in the lab to build silk scaffolding where they rested and at times fed (RE Alfaro, 2017, unpublished data). It is possible that this difference in silk use and foraging strategy between *H. carolinensis* and *D. tenebrosus*, especially the lack of regular web building, could account for the trends we observed with AC gland spigot numbers. Both species were observed on a few occasions to wrap prey items after a preliminary bite before returning to bite again. This is an ancient behavior seen in many other taxa including the Araneoidea, Phyxelididae, and other members of the RTA clade. It is also possible that they have evolved alternative uses for aciniform silk or do not need to produce this silk until the adult instars when numbers of spigots on both spinnerets increase.

Sexual dimorphism was observed in PI gland and AC gland spigots in *D. tenebrosus* and this has also been recorded in other lycosoids and araneid spiders (*Wąsowska, 1977*; *Townley & Tillinghast, 2009*; *Dolejš et al., 2014*). We were not able to rear our *H. carolinensis* males to adulthood in the lab colony, but given the trends in other Lycosidae, we would expect to observe sexual dimorphism as well (*Dolejš et al., 2014*). In most spiders, regardless

of lineage or cribellate or ecribellate status, adult males lose (abort) spigots of all types in the final molt: this is likely due to the shift in life history strategy of abandoning webs or territories to actively forage and look for females (*Wąsowska, 1977*; *Yu & Coddington, 1990*; *Hajer, 1991*; *Townley & Tillinghast, 2009*; *Dolejš et al., 2014*; R Carlson & CE Griswold, 1996, unpublished data; RE Alfaro, 2017, unpublished data).

Adult females of both species possessed a potential modified spigot (MS) on the PLS. A modified spigot was not reported for other lycosoids in previous studies (*Wąsowska, 1977*; *Dolejš et al., 2014*). Modified spigots have been reported as singular or, in some cases of cribellate silk users among members of the RTA clade, which includes lycosoids, the MS may occur with flanking spigots (*Griswold et al., 2005*; RE Alfaro, 2017, unpublished data). It is possible that this modified spigot in lycosoids could be homologous to the pseudoflagelliform gland spigot (PF) observed in cribellate lineages sister to the RTA clade, e.g., *Phyxelida tanganensis* (E. Simon & L. Fage, 1922) and *Hyptiotes paradoxus* (C.L. Koch, 1834) (Table S1; *Peters, 1984*; *Peters, 1995*; *Griswold et al., 2005*; *Eberhard, 2010*; *Eberhard & Barrantes, 2015*; R Carlson & CE Griswold, 1996, unpublished data). In contrast, it should be noted that in species with modified spigots, adult males typically possess an MS nubbin, which we did not observe in male *D. tenebrosus* (*Ramírez, 2014*, Table 5). To confirm these observations in *D. tenebrosus* and *H. carolinensis*, replicate adult female and male specimens are necessary.

## Phylogenetic comparative analyses

Six analyses had significant correlation coefficients after correcting for the bias of shared evolutionary history, suggesting that Instar, Strategy, and Type (spigot variety) are good predictors of spigot number in spiders (Table 5). Although most of the PGLS analyses found no significant correlation of the five independent variables with the 12 dependent variables of spigot numbers, this is not altogether surprising considering the PGLS analyses remove the bias of correlation due to shared evolutionary history. It is possible that with broader taxon sampling deeper in the phylogeny, trends may emerge beyond those explained by phylogenetic signal. As our analyses currently cannot include as a variable the full ontogeny dataset per species, it is also more likely with future developments of more complex statistical analyses within a phylogenetic context that will allow us to include the full ontogeny picture of each species, we will be able to gain a better understanding of what is driving silk spigot evolution in spiders.

Maximum number of instars (Instar) served as a proxy for body size or body condition in each spider species. Within a species with variation in maximum instar, we observed higher numbers of spigots in the individuals of older instars (Table S1). Finding a significant correlation between Instar and Average number of spigots in adult female aciniform and piriform gland spigots is not unexpected, considering these spigots increase in number with each instar (Table S1, Table 5). The more nutrition a juvenile spider consumes in one instar influences how much growth occurs in the molt to the next instar. Spiders with a steady food supply may invest in an increased number of instars to ensure better body condition at the adult stage, which could lead to a trend of increase in spigot numbers (RE Alfaro, 2017, unpublished data). Piriform gland spigot numbers were also positively

correlated with Type (variety of spigots possessed) (Table 5). Piriform gland silk is used as a cement for other silk fibers, particularly the major ampullate gland fibers in dragline silk (*Garb, 2013*). In orb weavers, piriform gland silk is used to cement the structural lines together and to the substrate whereas in wandering spiders it cements the dragline to the substrate to prevent the spider from falling (*Garb, 2013*). Orb weavers and lycosoids possess different types of silk spigots and have different uses for the shared spigots they possess (Table S1, Table 2); for example, MAP gland silk may be used to construct aerial frames (orb weavers) vs. surface dragline (lycosoids); aciniform gland silk may be used as web material in wolf spiders who spin funnel webs (the genus *Hipassa*) rather than as a prey wrapping material (orb weavers) (*Mathew, Sudhikumar & Sebastian, 2011*; *Garb, 2013*). It is possible that these differences in use are due to the types of spigots they possess and their differing foraging strategies (web building vs. predominately active hunting) and are what is causing this positive correlation we observed (Table 5). Finally, in adult female PGLS analyses, Strategy (web vs. non-web) was a significant predictor of the number of mAP gland spigots on the PMS. The coefficient of correlation was small, but when we look at the full ontogeny data (Table S1), we see a clear differentiation between araneoids and the others. In the adult female stage, araneoids lose one mAP gland spigot and retain one functional spigot, whereas in the other groups, from the lycosoids to the cribellate web builders, all female spiders retain the two mAP gland spigots that they possessed throughout their ontogeny (Table S1). The clear correlation between strategy as a predictor and mAP number as a response is expected since all araneoids possess one mAP spigot and the remaining spider groups possess two (Correlation coefficient: 0.291, $t = 2.448$, $p = 0.024$: Table 5). This may have withstood the phylogenetic correction of the PGLS, because several of the non-araneoids within our study also spin webs.

The PGLS analyses for second instars were largely non-significant. The lack of significant correlation between predictor and response variables may be due to second instars being more similar. Many spiders start out with the same general number of spigots upon emergence from the egg sac (second instar) and differentiation between foraging strategies may not be apparent at this instar, e.g., some second instar web builders do not spin webs (Table S1, *Hajer, 1991*; *Barrantes & Madrigal-Brenes, 2008*; RE Alfaro, 2017, unpublished data). However, both MAP gland spigots on the ALS and mAP gland spigots on the PMS were significantly correlated with Instar (maximum number of instars within each species) (Tables 2 and 5). This correlation with Instar is likely a case where having the full ontogeny incorporated into an analysis would provide clarity on this odd result. In general, web builders tended to have less number of instars to adulthood than non-web builders or wandering spiders; exceptions in this study are the ambush thomisid and sit & pursue (*Miller, Ament & Schmitz, 2014*) philodromid species: *Xysticus cristatus* (C. Clerck, 1757) and *Tibellus oblongus* (C.A. Walckenaer, 1802), respectively (Table S1), which had relatively few instars. Some second instars possess the two MAP gland spigots observed in all species later in ontogeny. However, both *X. cristatus* and *T. oblongus* had no MAP gland spigots in the second instar and *Metellina segmentata* (C. Clerck, 1757) possessed only 1 MAP gland spigot (Table S1; *Wąsowska, 1977*; *Yu & Coddington, 1990*; *Townley & Tillinghast, 2009*). These three species had some of the lower maximum numbers of instars

per species compared to the lycosoids, cribellate spiders and even viscous orb weavers (Table 2). Second instars across species possessed either none, one or two mAP gland spigots and this varied across foraging strategies and lineages. However, those species that possess both mAP gland spigots at the second instar were consistently the species with a higher maximum number of instars. This explains the significance of Instar as a predictor (Table S1).

## Ancestral character estimation

We also conducted an ancestral character estimation on specific spigots on the posterior lateral spinneret whose potential homology have long been debated (Fig. 3; *Peters, 1984*; *Peters, 1995*; *Griswold et al., 2005*; *Eberhard, 2010*; *Eberhard & Barrantes, 2015*; R Carlson & CE Griswold, 1996, unpublished data; RE Alfaro, 2017, unpublished data). We added an additional five species, including two species sister to the Araneoids, and three cribellate species ancestral to both the Araneoidea and RTA clades (Fig. 3). As we previously described we constrained our rate matrix (Table 6) that we used to model substitution rates across branch lengths based on prior knowledge about historical possession of the flagelliform, pseudoflagelliform and modified spigots in each taxon and considering Dollo's Law (Table 6) (*Dollo, 1893*; *Platnick & Griswold, 1991*; *Griswold et al., 2005*; *Dimitrov et al., 2016*; *Wheeler et al., 2016*).

The ancestral root of our phylogram of 27 species was more likely to have borne a modified spigot or none at all (likelihood: 0.365, 0.635, respectively) (Fig. 3, Results S5). Because this is not a full determination of the likelihood of a modified spigot or no spigot we cannot definitively determine the character state of the orb weaving ancestor at the node where Nicodamoidea + Araneoidea and the RTA clade split off (Fig. 3, Modified spigot likelihood: 0.398, None likelihood 0.602). It was unlikely to have possessed a pseudoflagelliform gland spigot (Fig. 3, Pseudoflagelliform likelihood: 0.000). We can, therefore, hypothesize that pseudoflagelliform and modified spigots are homologous structures and that modified spigots in the RTA clade likely are retained structures like those found in the sister and ancestral cribellate clades (Fig. 3). We cannot rule out that flagelliform gland spigots arose independently from modified spigots and thus cannot infer homology between this spigot and the others. We do not know the functionality of the modified spigots observed in *D. tenebrosus* and *H. carolinensis*. This would be useful to explore in the future, as the functionality of modified spigots in cribellate members of the RTA clade is the same as the pseudoflagelliform gland spigot (RE Alfaro et al., 2017, unpublished data). It is possible we are observing an intermediate stage of the loss of the modified spigot in *D. tenebrosus* and *H. carolinensis*.

Our results may change if we can incorporate ontogeny into the ACE analysis. For example, mimetids, which were ranked as having no spigots, do possess vestigial PLS spigots in the early instars (Table S1, *Townley & Tillinghast, 2009*). It is also possible that we are observing in real time the loss of the flagelliform gland spigot in this araneoid lineage. *T. perfuga* possess primordial modified spigot and flankers in second instars prior to them constructing webs in the third instar where functional spigots exist where the pre-spigots had been (Table S1; RE Alfaro, 2017, unpublished data). In most species,

the final molt of the adult male leads to loss or nubbins (non-functional spigots) in all three: Modified, Pseudoflagelliform, Flagelliform (Table S1; *W̧asowska, 1977*; *Yu & Coddington, 1990*; *Hajer, 1991*; *Townley & Tillinghast, 2009*; *Dolejš et al., 2014*; R Carlson & CE Griswold, 1996, unpublished data). This coincides with the male abandonment of the web for an alternative lifestyle of wandering in order to find females.

Future work, as the methods of phylogenetic inference grow and progress, may yield different results if we incorporate this analysis with the triad of spigots, i.e., the triplet of MS plus flankers, PF and flankers and/or FL (flagelliform) plus AG (aggregate), or spigot associations, i.e., PC with the PF in cribellate orb weavers. In araneoids, the flagelliform gland spigot is flanked by two aggregate gland spigots and in *T. perfuga* the modified spigot is flanked by two spigots of unknown gland association. In *Tengella,* these flankers resemble AC gland spigots but in some other cribellate spiders, e.g., *Matachia* or *Badumna* (*Griswold et al., 2005*; figs 87 A, D) the flankers resemble paracribellar (PC) gland spigots. The color-coded ACE phylogram shows a trend of loss (black color) for all spigot types in approximately half of the tip species (Fig. 3). This coincides with a shift in foraging strategies, i.e., from webs to running, observed in these lineages (Table 2) and is consistent with the current hypothesis that an adaptive tradeoff between silk production and fecundity is driving spider evolution to foraging strategies that do not involve silk or web building (*Blackledge, Coddington & Agnarsson, 2009*). Given the recent conclusions of phylogenomics studies indicating a much more ancient orb weaving ancestor and the new sister relationship of cribellate orb weavers to the RTA clade, our ACE results do indicate that deeper and broader sampling across the spider tree of life is necessary (*Bond et al., 2014*; *Fernández, Hormiga & Giribet, 2014*; *Garrison et al., 2016*).

Increased taxon sampling would also improve the ancestral character estimation analyses performed for the independent variables of the PGLS analyses. Not surprisingly, given the results and discussion of the above analysis, the ancestral state at the root node of the phylogram of 27 species was Web builder for Strategy (foraging strategy; likelihood = 1.0), and specifically either funnel (likelihood = 0.812), sheet (likelihood = 0.084) or orb webs (likelihood = 0.105) (Figs. 9 and 10). Also, given the ancestral state for the singular spigot on the PLS as Modified spigot (MS), one may reason that the ancestor was also a cribellate silk user. We found this to be the case after performing ACE on the independent variable Silk (main type of silk expressed), and found the ancestral state at the root of the phylogram was 100% likely to be cribellate (Fig. 11). Finally, Type or variety of spigot types, was less conclusive and we found likelihood values for all states at the root node (Table S2). However, the most likely ancestral states were those possessing a cribellum, paracribellum (likelihood = 0.398) and possibly a pseudoflagelliform gland spigot (likelihood = 0.433) (Fig. 12). This may not actually conflict with our results for the ACE of the singular PLS spigot, i.e., that MS is ancestral, in that we determined that the MS and PF spigots are homologous structures. But we also found that the most likely state was no modified spigot (likelihood = 0.635), which would suggest the plesiomorphic state of cribellum + paracribellum, without any modifications of spigots on the PLS is the ancestral state at the root node of the ultrametric phylogram.

As the techniques for more complex phylogenetic comparative analyses improve, such as to allow for a time-series dataset with multiple values per species, we suspect that incorporating the entire picture of spigot ontogeny will lead to some interesting and novel inferences about silk evolution. By not incorporating the entire ontogeny, but "snapshots" of the adult female and second instars, important observations are missed by the analyses, such as loss and regain of AC gland spigots on the PLS in *H. carolinensis,* or presence of FL-AG triad spigots in early Mimetidae instars. This approach to understanding spigot ontogeny from a phylogenetic comparative perspective is novel and we can only build on our efforts from this study by growing the dataset to include deeper taxon sampling and working towards the capability of phylogenetic statistical analyses that can function to accommodate ontogeny datasets as whole units for each species.

## CONCLUSIONS

In this study we explored spigot ontogeny in 22 species, including novel observations of *Dolomedes tenebrosus* and *Hogna carolinensis*. This is the first effort to create a phylogenetic comparative approach utilizing the recent Araneae Tree of Life. We performed 60 PGLS analyses of five independent variables: Strategy, Specific, Silk Type, Instar and twelve dependent variables (spigot numbers in adult females and second instars juveniles). Six had significant correlation coefficients indicating Instar, Strategy and Type (spigot variety) as good predictors of spigot numbers in spiders. Next, after adding five additional spider taxa to allow for deeper and broader taxon sampling within the Araneomorphae, we reconstructed ancestral character estimations of the unique, singular silk spigots on the PLS whose potential homology has been debated. The analysis predicted the ancestral root to either have no singular spigot or to possess a modified spigot (MS). Finally, additional ACE analyses on four of the five independent variables, suggest that the ancestral root of the RTA clade and Araneoidea was likely a spider that was cribellate, spun a web, and possessed a more diverse array of spigots. We also suggest that the modified spigot (MS) and pseudoflagelliform gland spigots (PF) are homologous structures. Current statistical methods do not allow for multiple values for species within PGLS, which limited our ontogenetic scope. However, as methods allow for more complex datasets, by looking at the full picture of spigot morphology during a spider life cycle across multiple clades of the AToL, we should be able to gain more depth of our understanding of silk evolution in spiders.

## ACKNOWLEDGEMENTS

The authors would like to thank Darrell Ubick and Erika Garcia for their assistance at CAS with SEM specimen preparation and imaging of *Dolomedes tenebrosus* and *Hogna carolinensis*. We wish to thank David Lightfoot and Kari Benson for donating the found females used to start the lab colonies of each species, and Sami Cordova for collecting two additional *H. carolinensis* females. We appreciate the conversations about spigot ontogeny of lycosids and the willingness to share data and observations by Petr Dolejš. We also acknowledge the work done by Robin Carlson to study the spigot ontogeny of the

spinning apparatus of *Phyxelida tanganensis*; her research was enabled by the CAS Summer Systematics Institute. We thank Sarah Goodacre, Martín J. Ramírez and Robert Raven for their thoughtful and constructive reviews of drafts of the manuscript.

### Funding

This work was supported by Graduate Resource Allocation Committee Research Grants from the Biology Graduate Student Association of University of New Mexico (RE Alfaro; $400, 2014; $400, 2015) and Alvin R. and Caroline G. Grove Summer Research Scholarships (RE Alfaro; $2000, 2015; $2225, 2016) through the Department of Biology, University of New Mexico. The unpublished study of spigot ontogeny of Phyxelida tanganensis by Robin Carlson and Charles Griswold was funded by NSF grant BIR-9531307. There was no additional external funding received for this study. The funders had no role in study design, data collection and analysis, decision to publish, or preparation of the manuscript.

### Grant Disclosures

The following grant information was disclosed by the authors:
Biology Graduate Student Association of University of New Mexico: $400, 2014, $400, 2015.
Department of Biology, University of New Mexico: $2000, 2015, $2225, 2016.
NSF grant: BIR-9531307.

### Competing Interests

The authors declare there are no competing interests.

### Author Contributions

- Rachael E. Alfaro conceived and designed the experiments, performed the experiments, analyzed the data, contributed reagents/materials/analysis tools, wrote the paper, prepared figures and/or tables, reviewed drafts of the paper.
- Charles E. Griswold conceived and designed the experiments, performed the experiments, contributed reagents/materials/analysis tools, wrote the paper, reviewed drafts of the paper.
- Kelly B. Miller conceived and designed the experiments, contributed reagents/materials/analysis tools, wrote the paper, reviewed drafts of the paper.

### Data Availability

   The raw data and R codes have been provided in the Supplemental Files.

### Supplemental Information

Supplemental information for this article can be found online at http://dx.doi.org/10.7717/peerj.4233#supplemental-information.

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
