# Peer review of "­Comparative spigot ontogeny across the spider tree of life"

_PeerJ, doi:10.7717/peerj.4233_

## Round 0.1 · original submission · Major Revisions

· Academic Editor

Major Revisions

Dear Dr. Alfaro,

Your manuscript has been reviewed carefully by two reviewers. I agree very much with the first reviewer, who emphasises the importance of the science that you present here. I also agree with their suggestion that the manunscript would benefit from some restructuring. I encourage you to consider incorporating the range of suggestions made regarding how to present and discuss your findings.

·

Basic reporting

The language is a bit verbose and very passive. The mss conveys a LOT of detail which is very important but diluted by the excesses. The following are examples: it is interesting that; e.g., we found this to be the case; we also found that the most likely state; it is possible that; we determined that; it is possible that; This means that; it would be helpful; we found that; we do know that; it is possible; etc. etc.

Recheck spelling of all family names. The addition of the initials of a species author seemed excessive and if it is a journal demand then it is mostly unnecessary. In only few cases are the author's initial needed, eg. L. Koch vs C.L. Koch, O.P-Cambridge vbs F.O.P, Cambridge. In a number of cases, e.g. line 420, 525, references are needed. Unless the information is provided new here, cite a source.
Also, ease off on the conjecture, e.g. line 524, 574.

Experimental design

The science is EXCELLENT; the depth and rigour of the analyses is more than adequate and the questions posed/tested great! This is a very important work that MUST be published.
Be careful with new "terms", like flankers and webless. Either put it in quotes are explain it. Please think about people for whom English is NOT their first language, e.g. l. 425 "hold steady" won't translate.
line. 517, must end in ', respectively"
Also, "sit and wait" works but "sit and pursue" seems contradictory, are these standard, if so, cite the original usage.l. 451. [We also observed snip!]... on PLS of both species" Which sexes?

Validity of the findings

The findings are totally consistent, very valuable. However, the conclusions are useless, they should resemble something like the abstract.

Additional comments

Really, it could do with a powerful tightening of language and probably a better structure of presentation of that amazing and valuable information you have.

·

Basic reporting

The article is well written in general, the references and context are well covered.

I think the title is a bit “overscoping” - it is supposedly about ontogeny in spiders, but the ontogenetic content is presented on a specific clade of Araneomorphae (Entelegynae). I know that journals and authors usually benefit from that :)

Experimental design

295 al., 2016). We used a maximum likelihood method with a model of the weighted rate matrix of
296 substitutions for these spigots (Table 6).
-- How was made the distinction between "absent" and "loss" in the scored species (data)?

Rates in Table 6 need an explanation. (Same with Table s2). It seems very strange that only a few scattered and rather derived taxa lack any MS-Fl-PsF, and yet the ancestral state is inferred as "absent". This probably comes from the rate matrix, which needs a justification. (Otherwise the interpretation of lines 388-390 is dubious.)

Validity of the findings

Figure 4: Where the identification of Pseudoflagelliform for Phyxelida comes from? I suspect it is arbitrary, since no histology was made (all were labeled as MS in Griswold et al. 2005).

384 gland spigots on the PLS (Table S1, Townley & Tillinghast, 2009). The predominately green
385 (Loss of MS spigot) clade was the RTA clade,
-- but green = "MS"

386 blue/green (Pseudoflagelliform or Modified, respectively)
-- but blue = "none"


340 ... In both species, a larger spigot was tentatively
341 identified as a ‘modified spigot’ (see Griswold et al., 2005: 61; character 96), with a potential
342 pre-modified spigot observed in the penultimate female stage of D. tenebrosus. These made no
343 other appearance in the ontogeny of both species spinning fields (Table 1, Figs. 7, 8).
-- This is strange, as there is no nubbin of a modified spigot (MS) in the male. According to my experience, MS spigots leave a nubbin in male (Ramirez 2014: Table 5).
The potential MS of Hogna is worth illustrating, since no MS was ever reported for lycosids.
This finding is later downplayed in the Discussion.

Additional comments

Minor edits:

73 2016). The former “Orbiculariae” Deinopoidea (cribellate orb builders) are now sister to the
74 RTA clade (includes wolf spiders and jumping spiders) rather than to the Araneoidea (sticky-silk
75 orb weavers) and Deinopoidea may not even be monophyletic (Garrison et al., 2016; Wheeler et
76 al., 2016).

-- I would say "closer" rather than "sister", because these relationships are unstable, and the Deinopoiea are not monophyletic, according to those studies.


Table 2 needs some explanation in the legend.


-- revise English:
99 & Opell, 2002; Blackledge et al., 2009; Pechmann et al., 2010). It is possible the higher
100 fecundity trends observed in orb-weavers and non-web builders compared to cribellate silk users

-- revise English:
163 taxa, particularly the singular, fiber producing spigot (MS, FL, PF) the on the PLS?




107 ... Most
108 of the Araneomorphae spiders possesses five types of spigots with another two appearing in
...
111 ... These are ...
116 ... 4) aciniform gland spigots (AC) on the PMS and 5)
117 aciniform gland spigots on the PLS that produce silk used in prey wrapping and lining egg sacs,
118 as well as the sheet portions in non-orb webs; and 6) cylindrical (=tubuliform) gland spigots
119 (CY) on the PMS and 7) cylindrical gland spigots on the PLS which are female specific and
120 produce fibers that form the egg sac (Fig.1, and see Fig. 1 in Garb, 2013).
-- items (4-5) and (6-7) sound like of the same type.


130 paracribellar spigots on the PMS. However, in T. perfuga, the modified spigot is flanked by two
131 smaller, unknown spigots whose function is currently undetermined (R.E. Alfaro, unpublished
132 data).
-- you can use Griswold et al. (2005) as a reference for this in T. radiata and many others.


291 N.I. Platnick (1987; Hypochilidae), Kukulcania hibernalis (N.M. Hentz, 1842; Filistadidae),
-- Filistatidae
292 Thaida peculiaris F. Karsch (1880),
-- ... (1880, Austrochilidae),

292 ... Megadictyna thilenii F. Dahl (1906; Nicodamidae)
-- Megadictynidae
293 ... The Nicodamidae are sister
-- Nicodamoidea

---

## Round 0.2 · accepted · Accept

· Academic Editor

Accept

The manuscript presents findings that allow us new insights into the evolution of particular morphological structures in spiders. The analyses and discussion are robust, clearly explained and thoughtful.

·

Basic reporting

The english is excellent, the references complete correct and through, and figures appropriately labelled. I had no complaint with the original submission but this is slightly improved.

Experimental design

The scope and aims are well defined and answered. The methods are well designed and thoroughly documented. Excellent work!

Validity of the findings

Although the results may not be earthshaking to some, to me they are really valuable and open new paths for future investigation. It is truly great work! Congratulations to the authors!